# Toxigenic Potential of Mesophilic and Psychrotolerant *Bacillus cereus* Isolates from Chilled Tofu

**DOI:** 10.3390/foods11121674

**Published:** 2022-06-07

**Authors:** Kyung-Min Park, Hyun-Jung Kim, Kee-Jai Park, Minseon Koo

**Affiliations:** 1Department of Food Safety and Distribution Research Group, Korea Research Institute, Wanju-gun 55365, Korea; parkkyungmin@kfri.re.kr (K.-M.P.); hjkim@kfri.re.kr (H.-J.K.); jake@kfri.re.kr (K.-J.P.); 2Department of Food Biotechnology, Korea University of Science & Technology, Daejeon 34113, Korea

**Keywords:** chilled tofu, *Bacillus cereus*, enterotoxin gene, antibiogram, biofilm formation

## Abstract

The prevalence, toxin gene profile, antibiogram, and biofilm formation to determine the virulence potential of mesophilic and psychrotolerant *Bacillus cereus* (*B. cereus*) isolated from chilled tofu were investigated. Among 58 isolates, 21 isolates were capable of growth at 7 °C, and these isolates shared a potential hazard for food poisoning with mesophilic isolates. *B. cereus* harboring enterotoxin genes was more frequently found in psychrotolerant isolates than in mesophilic isolates. Thirty-seven (62.2%) mesophilic isolates and all psychrotolerant isolates carried four or more enterotoxin genes. The hemolysin BL (42.9%) and nonhemolytic enterotoxin complexes (90.5%) were found at a higher frequency in psychrotolerant isolates than in mesophilic isolates. Some *B. cereus* isolates showed resistance to rifampicin or clindamycin, regardless of mesophilic and psychrotolerant isolates. A total of 56% and 40% mesophilic isolates displayed the strongest biofilm formation at 40 and 42 °C, respectively. However, the biofilm formation of psychrotolerant isolates was not significantly affected by temperature. The results of this study provide new strategies for the development of bacterial control, which allows us to optimize technologies to inhibit *B. cereus*, including psychrotolerant isolates, in the food industry.

## 1. Introduction

Tofu, a soft soybean curd, is known to be a low-calorie food and an excellent source of protein, with all nine essential amino acids, iron, calcium, and other micro-nutrients [1]. The consumption of tofu has increased due to these health benefits and the increasing popularity of vegetarian and vegan diets; therefore, worldwide tofu consumption is expected to rise in the coming years. However, tofu may provide a favorable condition for bacterial growth due to its high water content, high carbohydrate content, high protein content, and neutral pH [2]. Between 2000 and 2020, at least five tofu-associated foodborne outbreaks were reported to the National Outbreak Reporting System (NORS) by CDC, causing 140 illnesses and 5 hospitalizations as a result of *Listeria monocytogens* (*L. monocytogenes*) contamination [3]. Tofu has been linked to outbreaks involving psychrotolerant foodborne pathogens, including *Yersinia enterocolitica* and shigellosis occurring in commercial tofu [4,5]. In addition, *Staphylococcus aureus* (*S. aureus*) and Enterobacteriaceae have been found in commercial tofu [6,7], and unpackaged tofu was contaminated with various microorganisms, such as *Escherichia coli* (*E. coli*), *Salmonella* spp., and *Enterococcus* spp., in Thailand [8]. Although various foodborne pathogens were contaminated in tofu, *Bacillus cereus* (*B. cereus*) showed a relatively high incidence in both packed and unpacked tofu [8,9].

*B. cereus* is a Gram-positive, spore-forming, facultative anaerobe that is able to cause foodborne disease. *B. cereus* is responsible for two types of gastrointestinal diseases, emetic and diarrheal. Cereulide toxin as the causative factor of emetic illness is pre-formed during the vegetative growth of *B. cereus* in foodstuffs, and symptoms of vomiting occur within 1–5 h after the ingestion of the contaminated food with emetic toxins [10]. Enterotoxins, cytotoxin K (CytK), nonhemolytic enterotoxins (Nhes), hemolysin BL (Hbl), and Enterotoxin FM (EntFM) have been reported as causative factors for diarrheal cases from *B. cereus* [11]. The Nhe complex is encoded by *nheA*, *nheB*, and *nheC*, and the Hbl complex is composed of *hblA*, *hblC*, and *hblD* [12]. CytK toxins induce diarrhea symptoms via Nhe and Hbl toxins [13]. Some *B. cereus* strains can grow under 10 °C; thus, psychrotolerant *B. cereus* species are able to proliferate during cold storage and in cold chain conditions [14]. As psychrotolerant *B. cereus* species also produce hemolysin BL (HBL), nonhemolytic enterotoxins (NHEs), and emetic toxins, members of the psychrotolerant *B. cereus* species can affect microbiological and food quality [15]. Therefore, psychrotolerant *B. cereus* species should be considered a potential food poisoning hazard during food processing, distribution, storage, and cooking under low-temperature conditions.

Chilled food such as tofu, to ensure good quality and microbiological safety, are stored at refrigeration temperatures (at or below 8 °C) during their shelf life, but cold-tolerant bacteria, such as psychrotolerant *B. cereus* species, can proliferate in tofu during refrigerated storage. Due to their potential for bacterial growth at low temperatures, enterotoxin production, antibiotic resistance, and their implications for foodborne illness, *B. cereus* species are a concern in the food industry. Furthermore, *B. cereus* easily adsorbed and formed heterogeneous biofilm on food or on food contact surfaces [16]. The biofilm formation by *B. cereus* can originate from vegetative cells or from spores that become attached to surfaces, and the efficiency of sporulation is affected by stressful conditions. However, sporulation requires previous growth of vegetative cells and takes some hours to achieve; therefore, sporulation and germination of *B. cereus* are limited during food processing [16]. Biofilm cells are more highly resistant to stressful environmental conditions, including sanitizing or antibiotic treatment [17]. Since biofilm can form on surfaces related to the storage, processing, cooking, and serving of food products, biofilm-forming bacteria can cause serious public health problems; therefore, bacterial biofilm formation is currently an important topic in medical, environmental, and food microbiology [18]. The impact of different temperature conditions on biofilm formation has been studied for various foodborne pathogens, such as *Enterococcus* spp. [19], *Salmonella* spp. [20], and *L. monocytogenes* [21]. All studies indicate that temperature is a common factor in bacterial growth or survival and biofilm formation is modulated by temperature.

Microbiological investigation regarding tofu has consisted of attempts to determine microbiological quality, and several studies have shown that *B. cereus* is a significant food safety hazard in tofu samples [8,9,22]. However, few reports are available on the identification and toxigenic affiliation, including the toxin gene distribution, antibiogram, and biofilm-forming capability, of *B. cereus* isolated from tofu. Moreover, studies on behavioral characteristics, such as growth properties and biofilm formation at various temperatures, in *B. cereus* isolates from tofu are lacking. Therefore, the objectives of this study were: (1) to determine the growth properties of *B. cereus* isolates from tofu at temperatures lower or higher than the optimal temperature, (2) to investigate the possibility of toxigenic potential in *B. cereus* isolates, and (3) to compare the biofilm-forming capability of *B. cereus* isolates at various temperatures considering their growth temperature range.

## 2. Materials and Methods

### 2.1. Sample Preparation

A total of 20 packaged tofu produced in small- and medium-sized enterprises (SMEs) were collected from local supermarkets in Jeonju, Korea. Each product was twice; therefore, a total of 40 samples were analyzed. All samples were transported to the laboratory under refrigerated conditions within 1 h, and then the microbiological analysis was performed. Microbiological analysis for tofu was performed according to the Korea Food Code [23].

### 2.2. Enumeration of Aerobic Bacteria, E. coli, Coliforms, and B. cereus

First, 25 g of tofu was homogenized with 225 mL of 0.85% sterile saline for 2 min at 160 rpm using stomacher (Laboratory Blender Stomacher 400: Seward, London, UK). Then, ten-fold serial dilutions were performed with 9 mL of 0.85% sterile saline, and 1 mL of each dilution suspension was spread onto selective media for each microorganism. Aerobic bacteria, *E. coli*, and coliform were counted using a Petriflm^TM^ plate (3M microbiology, St Paul, MN, USA) for total aerobic bacteria and Sanita-kun plates (Sanita-Kun, Tokyo, Japan) for *E. coli* and coliform. Mannitol egg yolk polymyxin agar (MYP, Merck, Darmstadt, Germany) was used as the selective media for *B. cereus*. The plates were incubated at 37 °C for aerobic bacteria, *E. coli*, and coliform and 30 °C for *B. cereus* for 24 h (*E. coli*, coliform, *B. cereus*) or 48 h (aerobic bacteria). After incubation, the plates had 30–300 colonies, and specific colors for each microorganism were used for counting. Suspected colonies appearing as dry, pinkish colonies surrounded by a pink ring for *B. cereus* were selected, and a maximum of five colonies from each sample were subcultured on tryptic soy agar (TSA, Merck, Darmstadt, Germany). When there were less than five colonies, all were isolated. Biochemical identification of the selected colonies was conducted using the VITEX MS system (BioM’erieux, Inc., Marcyl’Etoile, France), according to the manufacturer’s directions. Total viable aerobic bacteria, *E. coli*, coliform, and *B. cereus* counts are expressed as colony-forming units per gram (CFU/g).

### 2.3. Bacterial DNA Extraction

To obtain the genomic DNA, 1 mL of overnight bacterial culture was centrifuged (15,000× *g*, 5 min), and the pellet was washed twice with 1 mL of phosphate-buffered saline (GibcoTM PBS, Thermo Fisher Scientific, Waltham, MA, USA). The cell pellet was resuspended in 100 µL of sterilized distilled water, heated at 95 °C for 10 min and then placed on ice. After centrifugation at 14,000× *g* for 2 min, the supernatant was used as the template.

### 2.4. Growth Properties

To determine the psychrotolerant properties, all *B. cereus* isolates were confirmed to have growth ability at 5, 7, 10, 40, 42, 45, 50, 52, and 55 °C. The *B. cereus* isolates were first inoculated on tryptic soy agar (TSA, Merck, Darmstadt, Germany) and incubated at 30 °C for 18 h. One colony of overnight culture was transferred on a fresh TSA plate and incubated at 5 °C for 40 days, at 7 °C for 25 days, at 10 °C for 10 days, and at 40, 42, 45, 50, 52, and 55 °C for 24 h. After incubation, growth properties were confirmed by presence of observable colonies on plate.

### 2.5. Detection of Enterotoxin and Emetic Toxin Genes

To analyze the pathogenic potential of the *B. cereus* isolates, all isolates were screened for the distribution of toxin genes encoding enterotoxins and emetic toxins by conventional PCR. The primer sequences for each toxin gene used in this study were based on our previous study [24,25] and are listed in Table 1. *B. cereus* ATCC 14579 (diarrheagenic) and NCCP 14796 (emetic) strains were used as the control strains.

### 2.6. Antibiotic Susceptibility Testing

Antibiotic susceptibility testing was performed by standard disk diffusion method on Mueller–Hinton agar (Merck, Germany) according to the Clinical and Laboratory Standards Institute [26].

The concentrations of antibiotic agents tested in this study were as follows: cefotaxime (30 µg), ceftriaxone (30 µg), chloramphenicol (30 µg), ciprofloxacin (5 µg), clindamycin (2 µg), erythromycin (15 µg), gentamicin (10 µg), imipenem (10 µg), penicillin (10 µg), rifampin (5 µg), tetracycline (30 µg), and vancomycin (10 µg). After incubation at 30 °C for 24 h, the antibiotic susceptibility was measured, and the results are expressed as “susceptible”, “intermediate”, and “resistant” [27].

### 2.7. Quantification of Biofilm Formation

The biofilm formation quantification of the *B. cereus* isolates was performed according to the modified method by Singh et al. [28] with different temperature conditions (7, 10, 30, 40, 42, 45, 47, 50, 52, and 55 °C) using the 96-well flat-bottomed polystyrene plates. To evaluate biofilm formation, all *B. cereus* isolates were inoculated into a tube containing 10 mL of tryptic soy broth (TSB, Merck, Germany) and were incubated for 18 h at 30 °C. A total of 10 µL of each bacterial suspension was inoculated into 190 µL of fresh TSB broth in each well of sterile 96-well plates. Negative control wells contained only TSB, and all tests were performed in triplicate. The plates were covered and incubated for 72 h at 30 °C, for 15 days at 10 °C, for 35 days at 7 °C, and 24 h at 40 to 55 °C. After incubation, the suspensions of all plates were poured off, and the wells were washed three times with 200 µL of phosphate-buffered saline. The attached bacteria were then fixed with 200 µL of methanol for 15 min, and methanol was removed from the plate and dried at room temperature. Then, the plates were stained with 100 µL of 0.5% crystal violet per well for 10 min; they were then washed with running tap water, and the plates were dried at room temperature. In this study, fresh TSB broth was tested as the negative control and *B. cereus* ATCC 10987 was used as the positive control strain. The optical density (OD) of each well was measured at 550 nm with a microplate reader (Synergy™ Mx, BioTek, Winooski, VT, USA). The isolates were classified as follows. The cut-off OD (ODc) was defined as three standard deviations above the mean OD value of the negative control. The strains were classified into four categories: no biofilm former (ODs ≤ ODc), weak biofilm former (ODc < ODs ≤ 2 × ODc), moderate biofilm former (2 × ODc < ODs ≤ 4 × ODc), and strong biofilm former (4 × ODc < ODs).

## 3. Results and Discussion

### 3.1. Microbial Contamination of Chilled Tofu

The contamination level of aerobic bacteria, *E. coli*, coliform, and *B. cereus* in packaged chilled tofu samples was examined. According to the results in Table 2, aerobic bacteria were detected in all the samples, and the mean value of the bacterial count was 3.5 ± 1.4 log CFU/g. The aerobic bacterial counts in most of the samples were distributed in 2 to 3 log CFU/g (55%), followed by 5–6 log CFU/g (20%), 4–5 log CFU/g (15%), and 3–4 log CFU/g (10%). Although there is no global legislation criterion for total aerobic bacteria in tofu and soybean products, food spoilage can occur with numbers of total aerobic bacteria between 10^6^ and 10^8^ CFU/g. As a result, the population of total aerobic bacteria in all tofu samples was acceptable for consumption.

The microbial quality of tofu may be affected by the handling conditions during the tofu manufacturing processing, including simmering soy milk, cooling and curdling the milk, and pressing the curd, pasteurizing, and packaging. Ribeiro et al. [29] reported that mesophilic bacteria in all tested tofu samples were 1–3 log CFU/g higher than those observed with soybean as the ingredient for tofu manufacturing. Insufficient washing of raw ingredients such as soybean to eliminate foodborne pathogens can lead to bacterial growth. In addition, improper cleaning or disinfection procedures for the processing line, filling or packing machines, plastic tray, or the cutting knife used during the tofu manufacturing process may induce a natural enhancement in the microbial load. Generally, packaged tofu is set in an anaerobic state, and pasteurization-cooling is added as an important processing step to improve the safety quality of tofu. Although packaging is a key element in tofu production as it prevents product deterioration and prolongs its shelf life, foodborne pathogens can still survive due to residual oxygen in the headspace because of inadequate evacuation or gas flushing, inappropriate packaging material leading to oxygen permeability, and poor sealing, resulting in small leakage. Some studies reported that coliform, *Pseudomonas* spp., *Enterococcus* spp., lactic acid bacteria, *B. cereus*, *S. aureus*, *Salmonella* spp., *Yersinia* spp., and *Cronobacter sakazakii* [8,30] were detected in tofu. In our study, coliform bacteria were detected in 12 samples, and 6 samples among coliform-positive samples had coliform levels of more than 4 log CFU/g. In Korea (Food code No. 2021-54), the microbiological criteria for soybean products including tofu are limited for coliform (*n* = 5, c = 1, m = 0, and M = 10; only applicable for filled and sealed-packaged products). According to standards and specifications for general foods of Korea, the count for *B. cereus* should not exceed more than 10^3^ CFU/g, and foodborne pathogens should be negative. Approximately 30% of tofu samples showed coliform counts above the legal limit of Korea guidelines, indicating the need for the improvement of manufacturing and hygiene practices during tofu processing. When *B. cereus* was isolated from commercial tofu in Korea and Chinese fermented tofu, the samples were contaminated with more than 5 log CFU/g of *B. cereus* [9,31]. For tofu products in America (FDA circular No. 2013-010), the limit is established for only *B. cereus* and *S. aureus* (satisfactory: lower than 10^2^–10^3^ CFU/g). However, if the *B. cereus* level is over 10^5^ CFU/g, it is generally deemed that the food products need to be evaluated for potential hazards. In our study, 92.5% of samples had less than 3 log CFU/g, and *B. cereus* concentrations of up to 3 log CFU/g were observed in only three sample. Whereas *B. cereus* was not detected in the filling water of any tofu samples. Most large-scale factory-made tofu is produced under the hazard analysis and critical control point system, whereas tofu produced at a small industrial scale may be contaminated by the growth of pathogens during processing due to the lack of strict standards for quality control.

### 3.2. Growth Properties of B. cereus Isolates from Chilled Tofu

Chilled food is generally kept at 8 °C or below, and chilling is one of the most widely practiced methods of controlling microbial growth and maintaining food quality in tofu. However, considerable opportunities for the temperature fluctuation of chilled foods can arise during manufacturing, distribution, storage, and retail sale, as well as in the home, and temperature fluctuation has been associated with the increased proliferation of psychrotolerant bacteria [32]. Thus, monitoring the prevalence of psychrotolerant strains among *B. cereus* isolated from tofu could be an important food safety measure for predicting potential food safety hazards.

We tested the growth properties of 58 *B. cereus* isolates from tofu samples in different temperature conditions, and the results are shown in Table 3. It was observed that 36.2% of the 58 isolates grew at 7 °C, and more than 60% of the isolates could grow at 10 °C (65.5%). All isolates were capable of growth at 12 °C, in contrast to showing no growth at 5 °C. At a temperature up to 40 °C, 63.8% of isolates could grow at 47 °C. All tofu isolates were capable of growth at 40, 42, and 45 °C. In this study, the growth of approximately 36% of *B. cereus* isolates occurred at 7–45 °C for psychrotolerant strains, but not at 5 °C or up to 47 °C. The temperature range for 34.5% of isolates was between 12 and 47 °C, and 29.3% of isolates showed a temperature range from 10 to 47 °C. At more than 50 °C, none of the isolates showed any growth. 

After packaging, tofu is preserved and sold in retail markets or supermarkets under cold conditions. Therefore, inadequate temperature control can be a main cause for proliferation of psychrotolerant *B. cereus*. Our results suggest that tofu should be stored at a temperature lower than 5 °C to prevent the growth of *B. cereus*, including psychrotolerant strains. Psychrotolerant *B. cereus* is usually isolated in soil, milk, ice cream, dairy products, rice salad, spices, and vegetables [15,33,34], whereas *B. cereus* from chilled food containing vegetables did not grow at 10 °C [24]. Psychrotolerant *B. cereus* isolates can become a potential hazard of food poisoning due to their toxigenic characteristics. In a previous study, most psychrotolerant *B. cereus* isolates from lettuce contained various combinations of the enterotoxin gene [15]. Other studies reported that psychrotolerant *B. cereus* isolates carry *nheABC* and *hblACD*, which are associated with diarrheal disease [35,36]. Further studies may be necessary to evaluate the characteristics of the toxigenic gene distribution, antibiotic resistance, and biofilm formation of *B. cereus* isolates, including mesophilic and psychrotolerant isolates, to predict the possible occurrence of food poisoning.

### 3.3. Toxigenic Potential of Psychrotolerant and Mesophilic B. cereus Isolates from Chilled Tofu

The *hblA*, *hblC*, *hblD*, *nheA*, *nheB*, *nheC*, *cytK*, *entFM*, and *ces* are major genes associated with foodborne illness produced by *B. cereus* [37]. To confirm the food poisoning potential of *B. cereus* isolates from tofu, including psychrotolerant isolates, we investigated the toxin gene distribution and profile of 37 mesophilic and 21 psychrotolerant isolates. According to the results shown in Table 4, *B. cereus* harboring enterotoxin genes was more frequently found in psychrotolerant isolates than in mesophilic isolates. 

The analysis of the distribution of *nheA*, *nheB*, and *nheC* indicated that the *nheC* gene was the predominant enterotoxin gene, regardless of psychrotolerant and mesophilic isolates, with prevalence rates of 97.3% (mesophilic isolates) and 100% (psychrotolerant isolates). In addition, the detection rate of the NHE complex encoded by *nheA*, *nheB*, and *nheC* in psychrotolerant isolates was significantly higher (90.5%) than that in mesophilic isolates (67.6%). The detection rate of the HBL complex composed of *hblA*, *hblC*, and *hblD* showed a significant difference between psychrotolerant and mesophilic isolates. There were 42.9% psychrotolerant isolates with *hblA*, *hblC*, and *hblD*, while only 2.7% mesophilic isolates carried three *hbl* genes. The *entFM* gene, which might cause necrotic enteritis, was detected in 73% of the mesophilic isolates and 100% of the psychrotolerant isolates. The percentage of isolates carrying *entFM* is consistent with previous studies, which demonstrated that 90–100% of food-related *B. cereus* isolates possessed *entFM* [15,37,38]. The *cytK* gene was detected in 5.4% of mesophilic isolates and 23.8% of psychrotolerant isolates. Similar results were found for strains isolated from cooked chilled food [39], vegetables [39], meat products [40], and soybean products [38]. None of the isolates harbored the *ces* gene encoding cereulide. A total of 62% of mesophilic isolates and all psychrotolerant isolates possessed more than four enterotoxin genes. Previous studies reported that psychrotolerant *B. cereus* harbors *nheA*, *hblA*, and *cytK* with high hemolytic activity [33,35,36,41]. Although the occurrence of enterotoxin genes shows the difference between mesophilic and psychrotolerant isolates, tofu may cause a potential health risk due to the proliferation of mesophilic and psychrotolerant *B. cereus* harboring enterotoxin genes during long-term cold storage.

### 3.4. Antibiotic Susceptibility Test of Mesophilic and Psychrotolerant B. cereus Isolates from Chilled Tofu

The resistance patterns to 12 antibiotics in *B. cereus* isolates were evaluated, and the results are shown in Figure 1. No significant difference in the antibiotic susceptibility test was observed between mesophilic and psychrotolerant isolates. All isolates showed resistance to penicillin, ceftriaxone, and cefotaxime and were sensitive to imipenem, ciprofloxacin, tetracycline, gentamicin, vancomycin, and chloraphenicol.

Merzouqui et al. (2014) reported that *B. cereus* isolated from milk, dairy products, spices, and salad with rice was susceptible to chloramphenicol (67.2%) and erythromycin (84.4%), which is consistent with the results of our study. *B. cereus* is generally resistant to β-lactam antibiotics, including third-generation cephalosporins due to β-lactamase-producing bacteria. Rifampicin has a bactericidal effect against microorganisms due to catalyzing the transcription of DNA to RNA, but *B. cereus* is frequently resistant to rifampicin [25,42]. In our study, 10.8% of mesophilic isolates and 9.5% of psychrotolerant isolates showed resistance to rifampicin. These results are much lower than the rifampicin resistance rate of vegetables (49–83%), rice, and cereal (62%), but higher than those of different kinds of foods, including traditional dairy products (0%) and ready-to-eat foods (0%) [15,25,43,44,45]. Even though the incidence of multi-antibiotic-resistant bacteria was not present in either mesophilic or psychrotolerant isolates, it is still a matter of concern, since antibiotic-resistant isolates may survive during gastrointestinal passage and may transfer the antibiotic resistance genes or plasmid-encoded virulent genes to other antibiotic-susceptible bacteria [46].

### 3.5. Biofilm-Forming Capacity at Different Temperatures of Mesophilic and Psychrotolerant B. cereus Isolates from Chilled Tofu

Among various environmental factors affecting the bacterial biofilm, temperatures critically regulate the stages of biofilm formation [21]. Tofu samples can be exposed to temperature fluctuation during processing, storage, cooking, and serving, and the temperature change can affect the biofilm-forming capability of *B. cereus* isolates. We evaluated the impact of different temperatures, considering their growth temperature range, on the development of *B. cereus* biofilm formation. According to the results in Figure 2, all psychrotolerant *B. cereus* isolates showed non- or weak biofilm formers at the tested temperatures, while mesophilic *B. cereus* isolates showed a higher biofilm-forming ability at a relatively high temperature of 40–45 °C than at 7, 10, 47, and 50 °C.

Among mesophilic isolates at the optimal temperature (30 °C), 16.2% (six isolates) were strong biofilm formers and 29.7% of isolates showed moderate biofilm-forming activity. Interestingly, of the mesophilic isolates, at 40 °C, most of the isolates (86.5%) showed higher than moderate biofilm-forming ability; specifically, 21 isolates (56.8%) were strong biofilm formers. When the temperature was 42 °C, 40.5% of mesophilic isolates formed strong biofilms, and 27% of isolates were able to form a moderate biofilm. Furthermore, 10.8% of mesophilic isolates exhibited a moderate biofilm-forming capacity at 45 °C, in contrast to all mesophilic *B. cereus* isolates with non-biofilm formers at low temperatures. Tofu preparation involves a heating process, including the boiling of soymilk and pasteurization of packaged tofu. Therefore, *B. cereus* surviving in final products may show a greater tendency to form the biofilm at higher temperatures than their optimal temperature in comparison with low temperatures. Harimawan et al. showed that *Serratia marcescens* had greater biofilm-forming abilities at a high temperature (50 °C) compared with the optimal temperature (37 °C) [47]. Han et al. reported that *Vibrio parahaemolyticus* showed increased biofilm-forming capability with increasing temperature [48]. Rode et al. also indicated that the biofilm formation of *S. aureus* is strongest at 46 °C, rather than at the optimal temperature (37 °C) [49]. *Bacillus pumilus* also formed at 50 °C, with the highest attachment to a polystyrene plate [50]. The extreme temperature of 50 °C made *Bacillus haynesii* adhere better to the hydrophobic polystyrene surface and establish biofilm molecules [51]. In contrast to mesophilic isolates, biofilm development at both below and above the optimal temperature did not occur in psychrotolerant *B. cereus* isolates. These results are contrary to those reported by a previous study in which psychrotolerant isolates displayed greater biofilm-forming activity at 7 °C than at 10 °C or 30 °C [15]. We considered that the differences in biofilm-forming capabilities of psychrotolerant *B. cereus* isolates may be attributed to food ingredient/sources, their belonging to different food groups, or the food processing environment.

The exact mechanisms associated with increased biofilm formation at different temperatures are not known. However, the production of extracellular polymeric substances (EPSs), which are known to enhance the adherence capability of bacteria and attachment to food contact surfaces for biofilm formation, may be associated with biofilm development at increased temperatures [52]. In addition, Stepanovic et al. reported that the production of fimbriae related to the increased biofilm formation capability of *Salmonella* spp. at different temperatures [20], and Chavant et al. reported that temperature increased the cell hydrophilic properties of *L. monocytogenes* and altered the bacteria’s ability to adhere to hydrophobic materials, such as polystyrene [53]. Temperature regulates the expression of many genes related to stress adaptation in microorganisms by changing a cell surface charge that could affect biofilm formation [21]. The current study indicated that the biofilm formation of *B. cereus* isolates might be influenced by the growth environment, which indicates that the temperature fluctuation occurring during the processing, storage, cooking, and serving of tofu can form a strong biofilm on food or food contact surfaces by effecting physiological changes, such as EPS production, cell surface charge, and hydrophilic properties. Tofu is frequently used in schools, restaurants, and homes in boiled, fried, or steamed form, as it is a high-protein meat alternative. After food is cooked, hot food should be kept hot at 60 °C or warmer to prevent bacterial growth. If food items are left in a temperature danger zone of less than 50 °C, *B. cereus* isolates can promote bacterial growth and form biofilms on food or food contact surfaces and pose a potential health risk to susceptible consumers. *B. cereus* is able to form the biofilm on equipment and utensils, and exists either in a vegetative or in spore form that is highly resistant to antimicrobials and cleaning procedures [54]. Further study is necessary to understand the mechanism related to the change in the biofilm formation of *B. cereus* isolates according to temperature fluctuation.

## 4. Conclusions

We observed that some *B. cereus* isolates from chilled tofu could grow at 7 °C, and these psychrotolerant isolates share the virulence potential with mesophilic isolates. These data suggest that higher temperatures (40–45 °C) than the optimal temperature (30 °C) may enhance the biofilm formation of mesophilic *B. cereus* isolates. As *B. cereus* in tofu can induce biofilm formation through environmental change, it can enhance the food safety risk associated with tofu consumption and the risk of cross-contamination in tofu processing. The finding that biofilm formation may be promoted under changing temperature conditions of food and toxigenic psychrotolerant *B. cereus* present in chilled tofu indicates that the tofu industry should be aware that controlling the temperature of enterotoxigenic and biofilm-forming *B. cereus* isolates may be important. Tofu is often used as the main ingredient in salads and consumed in various forms, such as fried or steamed. Therefore, strict temperature and quality control to avoid the bacterial growth and biofilm formation of *B. cereus*, including psychrotolerant isolates, should be carefully managed throughout the tofu industry (processing, distribution, storage, cooking, and serving) for safe consumption.

## Figures and Tables

**Figure 1 foods-11-01674-f001:**
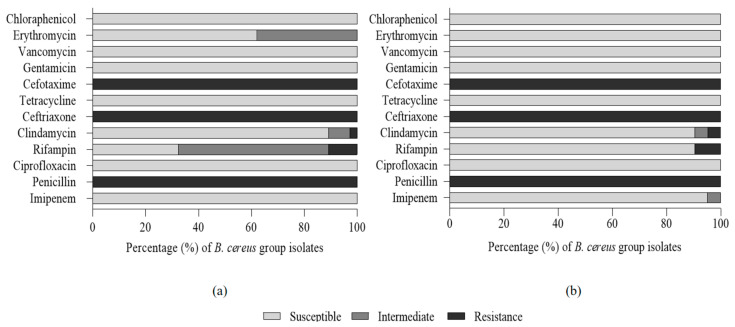
Antibiogram of mesophilic (**a**) and psychrotolerant (**b**) *B. cereus* isolates from chilled tofu.

**Figure 2 foods-11-01674-f002:**
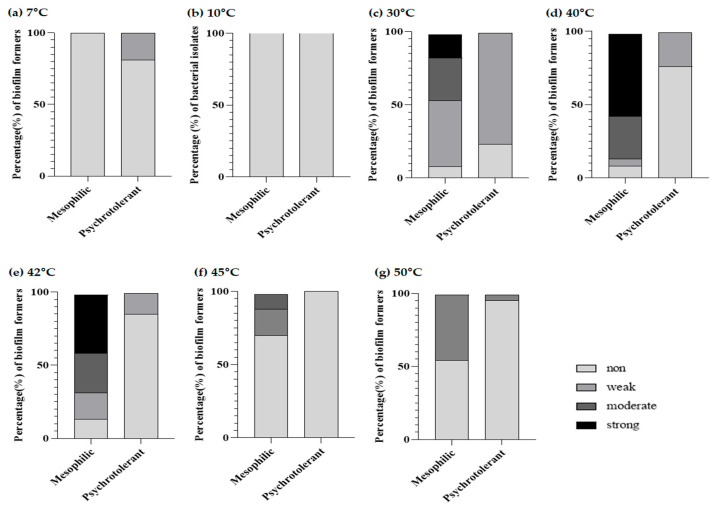
Biofilm phenotype (%) of mesophilic and psychrotolerant *B. cereus* isolates from chilled tofu at 7 °C (**a**), 10 °C (**b**), 30 °C (**c**), 40 °C (**d**), 42 °C (**e**), 45 °C (**f**), and 50 °C (**g**).

**Table 1 foods-11-01674-t001:** Sequences of primers used in this study.

Target Gene	Primer	Sequence (5′-3′)	Melting Temp (°C)	Amplicon (bp)
*hblA*	hblA-F	GTG CAG ATG TTG ATG CCG AT	55	319
	hblA-R	ATG CCA CTG CGT GGA CAT AT		
*hblC*	hblC-F	AAT GGT CAT CGG AAC TCT AT	55	749
	hblC-R	CTC GCT GTT CTG CTG TTA AT		
*hblD*	hblD-F	AAT CAA GAG CTG TCA CGA AT	55	429
	hblD-R	CAC CAA TTG ACC ATG CTA AT		
*nheA*	nheA-F	GTT TTT ATT GCT TCA TCG GCT	55	499
	nheA-R	CTA TCA GCA CTT ATG GCA G		
*nheB*	nheB-F	CTA TCA GCA CTT ATG GCA G	55	769
	nheB-R	ACT CCT AGC CGG TGT TCC		
*nheC*	nheC-F	CGG TAG TGA TTG CTG GG	55	581
	nheC-R	CAG CAT TCG TAC TTG CCA A		
*entFM*	entFM-F	ATG AAA AAA GTA ATT TGC AGG	55	1269
	entFM-R	TTA GTA TGC TTT TGT GTA ACC		
*cytK*	cytK-F	GTA ACT TTC ATT GAT GAT CC	44	505
	cytK-R	GAA TAC TAA ATA ATT GGT TTC C		
*ces*	ces-F	TTGTTGGAATTGTCGCAGAG	60	405
	ces-R	GTAAGCGAACCTGTCTGTAACAACA		

**Table 2 foods-11-01674-t002:** Microbial counts of total aerobic bacteria, coliform, *B. cereus*, and *E. coli* in chilled tofu.

Microorganism	No. (%) of Positive Samples	Population Range of Bacteria (log CFU/g)					
		0–1	1–2	2–3	3–4	4–5	5–6
Total aerobic bacteria	40/40 (100%)	0 (0%)	0 (0%)	22 (55%)	4 (10%)	6 (15%)	8 (20%)
Coliform	12/40 (30%)	28 (70%)	2 (5%)	4 (10%)	0 (0%)	2 (5%)	4 (10%)
*B. cereus*	6/40 (15%)	34 (85%)	0 (0%)	3 (7.5%)	2 (5%)	1 (2.5%)	0 (0%)
*E. coli*	0/40 (0%)	40 (100%)	0 (0%)	0 (0%)	0 (0%)	0 (0%)	0 (0%)

**Table 3 foods-11-01674-t003:** Evaluation of the growth ability of *B. cereus* isolates from chilled tofu at different temperatures.

Temperature for Growth (°C)	No. (%) of *B. cereus* Isolates
5	0 (0.0%)
7	21 (36.2%)
10	38 (65.5%)
12	58 (100%)
40	58 (100%)
42	58 (100%)
45	58 (100%)
47	37 (63.8%)
50	0 (0.0%)
52	0 (0.0%)
55	0 (0.0%)

**Table 4 foods-11-01674-t004:** Frequency and profile of toxigenic mesophilic and psychrotolerant *B. cereus* isolates from chilled tofu.

Toxin Gene	No. (%) of Toxigenic *B. cereus* Isolates	
	Mesophilic Isolates (*n* = 37)	Psychrotolerant Isolates (*n* = 21)
Frequency of toxin gene		
*nheA*	37 (100%)	19 (90.5%)
*nheB*	25 (67.6%)	21 (100%)
*nheC*	36 (97.3%)	21 (100%)
*nheABC*	25 (67.6%)	19 (90.5%)
*hblA*	1 (2.7%)	9 (42.9%)
*hblC*	7 (18.9%)	19 (90.5%)
*hblD*	6 (16.2%)	21 (100%)
*hblACD*	1 (2.7%)	9 (42.9%)
*entFM*	27 (73.0%)	21 (100%)
*cytK*	2 (5.4%)	5 (23.8%)
*ces*	0 (0.0%)	0 (0.0%)
Profile of enterotoxin gene		
*NheABC + hblACD + cytK + entFM*	1 (2.7%)	0 (0.0%)
*nheABC + hblACD + entFM*	0 (0.0%)	7 (33.3%)
*nheABC + cytK + entFM*	1 (2.7%)	5 (23.8%)
*nheABC + entFM*	21 (56.8%)	7 (33.3%)
*hblACD + entFM*	0 (0.0%)	2 (9.5%)
*nheA + nheC*	8 (21.6%)	0 (0.0%)
*nheABC*	2 (5.4%)	0 (0.0%)
*entFM*	4 (10.8%)	0 (0.0%)

## Data Availability

The data presented in this study are available within the article.

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
