# Peer review of "Toxigenic Potential of Mesophilic and Psychrotolerant Bacillus cereus Isolates from Chilled Tofu"

_foods, 2022, doi:10.3390/foods11121674_

Round 1

Reviewer 1 Report

Article entitled ‘Toxigenic Potential of Mesophilic and Psychrotolerant Bacillus  cereus Isolates from Chilled Tofu” shows results useful to the journal's readers.

Specific comments:

Introduction section:

Line 80-81 can be omitted once similar to line 30-31.

Some duplicated phrases (line 71 similar to 75) turn the reading boring. I suggest rewriting it.

Material and Methods:

Line 108-109 : MYP agar…… It seems some information is missing here

Line 192 : storage stored….

Line 307 :   “ polystyrene”

Results and Discussion:

Table 1.  Do the authors obtain the bacterial contamination profile of tofu samples accordingly to official standards regulation of food safety? I mean, are the samples adequate for consumption?  What are the contamination levels allowed for tofu in your country? This information can also be included in the text.

Do the authors observe the emergence of endospores? Especially at low temperatures, maybe the vegetative growth is not present but bacteria sporulate and survive. What is the impact of sporulation on biofilm formation at low temperatures? I suggest discussing this in the text.

Line 308-312: The lack of biofilm formation at low temperatures by psycrotolerant isolates should be discussed in more detail compared to previous work ( reference 15) . Why are the results the opposite? 

Author Response

# Reviewer 1

Dear reviewer, thank you very much for your kind judgement of our manuscript. We grateful for the time you expended to improve our work. In the following sections, you will find our responses to each of your points and suggestions.

Line 80-81 can be omitted once similar to line 30-31.

I deleted a similar sentence to line 30-31

Some duplicated phrases (line 71 similar to 75) turn the reading boring. I suggest rewriting it.

I deleted ‘Pathogen in food products experience temperature fluctuation during harvest, processing, distribution, storage, and cooking’ of Line 74

Line 108-109 : MYP agar…… It seems some information is missing here

I inserted the information about incubation time and temperature for MYP agar (Line 107-108). 

Line 192 : storage stored….

I revised the sentence like line 219.

Line 307 :   “ polystyrene”

I revised from ‘polysyrene’ to ‘polystyrene’ (Line 343).

Table 1.  Do the authors obtain the bacterial contamination profile of tofu samples accordingly to official standards regulation of food safety? I mean, are the samples adequate for consumption?  What are the contamination levels allowed for tofu in your country? This information can also be included in the text.

Thank you for your comments. We agree with the reviewer's suggestion.

Although there is no a legislation criterion for total aerobic bacteria for tofu and soybean products in the worldwide, food spoilage can occur with total aerobic bacteria of much higher between106-108CFU/g (Public Health Laboratory Service, UK). In our study, the level of total aerobic bacteria was distributed in 102-103 CFU/g (55%), followed by 105-106 CFU/g (20%), 104-105 CFU/g (15%), and 103-104 CFU/g (10%). For tofu products in America (FDA circular No. 2013-010), the limit is established for only B. cereus and S. aureus (satisfactory; lower than 102-103 CFU/g). In Korea (Food code No. 2021-54), the microbiological criteria for tofu and soybean products is established in coliform (n=5, c=1, m-0, M=10; only applicable to filled-and sealed-packaged products). According to standards and specifications for general foods of Korea, the count for B. cereus should not exceed more than 103 CFU/g and foodborne pathogens should be negative. In present study, 30% tofu samples showed coliform counts above the legal limit of Korea guideline and most of tofu samples showed the level of less than 103 CFU/g. B. cereus concentrations of up to 103 CFU/g were observed in three sample.

We included the above mentioned text in Results and discussion 3.1. Microbial contamination of chilled tofu (Line 176-212).

Do the authors observe the emergence of endospores? Especially at low temperatures, maybe the vegetative growth is not present but bacteria sporulate and survive. What is the impact of sporulation on biofilm formation at low temperatures? I suggest discussing this in the text.

Thank you for providing these insights about our manuscript.

Like reviewer's comments, the biofilm formation by B. cereus can originate from vegetative cells or from spores that become attached to surfaces and the efficiency of sporulation is affected by low temperature. Vegetative cells can be exposed to various environmental conditions such as low temperature during or after biofilm formation ant could stimulate sporulation of B. cereus within biofilm matrix. However, sporulation needs previous growth of vegetative cells and takes some hours to achieve. Consequently, environmental stress conditions for spore formation and germination by B. cereus are rarely encountered for long periods (Gauvry et al., 2017) and therefore sporulation and germination of B. cereus is limited during food processing. Moreover, Ryu et al. reported that the majority of cells in B. cereus biofilm at 12°C is in a vegetative form (Ryu et al., 2005). Sporulation, or biofilm formation for B. cereus type culture strains have been intensively studied, however, there is still lack of information on the role of biofilm formation by vegetative form of B. cereus food isolates and their correlation on biofilm formation and temperature. A greater understanding of the relationship between biofilm formation and temperature may be important to improve the bacterial inhibition technology in food manufacture. However, our further study will consider the possible correlation in sporulation and biofilm formation at environmental stress condition. 

The above mentioned text included in introduction section of manuscript (Line 65-70).

 Line 308-312: The lack of biofilm formation at low temperatures by psycrotolerant isolates should be discussed in more detail compared to previous work ( reference 15) . Why are the results the opposite?

Thank you for providing these insights about our manuscript.

In previous study, we collected the green leaf lettuce from farm-to-retail during cold chain. After harvest, leafy vegetables are packaged at low temperature condition and were loaded onto refrigerated trucks for transportation and storage in a distribution center and retail shop with cold condition. The temperature of leafy vegetables was maintained at 5-7 °C throughout the entire distribution chain for over 2 weeks. While, the tofu production process includes soybean pretreatment, washing, soaking, grinding, boiling, pulping, pressing, and packing without cold environment. After packaging, tofu is preserved and sold refrigerated in retail market and supermarkets under cold condition for shelf life extension. Tofu preparation involves heating process, including boiling of soymilk and pasteruization of packaged tofu. Therefore, B. cereus survived in final products may show a greater tendency to form the biofilm at higher temperatures than their optimal temperature in comparison with low temperature. As a result of different food processing environment and background microorganism characteristics of leafy vegetable and tofu, B. cereus isolates from leafy vegetables may be exposed to low temperature conditions for a longer time than those from tofu. Therefore B. cereus isolates from leafy vegetables under nutrient deficiency by distribution period for over 2 weeks may show resistance to low temperature with enhancement of biofilm formation ability to survival. We consider that the differences of biofilm forming capability by psychrotolerant B. cereus isolates may be attributed to food ingredient/soruces, their different food group, food processing environment.

The above mentioned text included in Line 347-349

Reviewer 2 Report

This manuscript deals with the growth properties, toxigenic potential and biofilm formation of Bacillus cereus (B. cereus) isolates from packaged chilled tofu samples. The presented results are very important for enhancing hygiene management in the tofu processing industry, schools, restaurants, and homes.

 (1) Did the authors check the presence of mesophilic and psychrotolerant B. cereus isolates in the filling water of the packed tofu? Did the authors evaluate the presence or the biofilm formation of mesophilic and psychrotolerant B. cereus isolates on the surface of the packaging material of 20 packaged tofu used in this study?

(2) It is necessary to consider and explain the contamination sources of mesophilic and psychrotolerant B. cereus isolates in the tofu manufacturing process.

(3) There are a number of misspellings and improper word usage in the text.

p. 1, Abstract, line 11, “B. cereus” should be “Bacillus cereus (B. cereus)”

p. 1, Abstract, line 22, “psychrotolerant and mesophilic” should be “mesophilic and psychrotolerant”

p. 1, line 30-33, “the consumption of tofu has increased; therefore,” should be deleted

p. 1, line 37, “Listeria monocytogens” should be “Listeria monocytogenes (L. monocytogenes)”.

p. 1, line 39, “[4-5]” should be “[4,5]”

p. 1, line 40, “Staphylococcus aureus” should be “Staphylococcus aureus (S. aureus)”

p. 1, line 40, “[6-7]” should be “[6,7]”

p. 1, line 42, “Escherichia coli” should be “Escherichia coli (E. coli)”

p. 1, line 43, “Bacillus cereus” should be “Bacillus cereus (B. cereus)”

p. 2, line 69, “Biofilm” should be “biofilm”

p. 2, line 80-81, “vegetarian vegan and hypocaloric diets” should be “vegetarian, vegan, and hypocaloric diets”

p. 3, line 99, “2.2. Microbiological quality of chilled tofu” should be “2.2. Microbiological Quality of Chilled Tofu

p. 3, line 109, The incubation temperature and time for MYP agar should be described.

p. 3, line 133, “[24-25]” should be “[24,25]”

p. 4, line 152-153, “10 µL of over-night culture” should be “An over-night culture of 10 µL”

p. 4, line 163, “3.1. Microbiological quality of Chilled Tofu” should be “3.1. Microbiological Quality of Chilled Tofu

p. 4, line 171-172, the caption of Table 1, “Bacillus cereus, and Escherichia coli in chilled tofu” should be “B. cereus, and E. coli in chilled tofu”

p. 4, Table 1, column 1, row 4, “Bacillus cereus” should be “B. cereus

p. 4, Table 1, column 1, row 5, “Escherichia coli” should be “E. coli

p. 5, line 180, “Staphylococcus aureus” should be “S. aureus

p. 5, line 208, the caption of Table 2, “Bacillus cereus” should be “B. cereus

p. 5, Table 2, column 2, row 1, “Bacillus cereus” should be “B. cereus

p. 5, line 180, “could grow at 47 °C, and.” should be “could grow at 47 °C.

p. 6, line 210, “milk, and ice cream” should be “milk and ice cream

p. 6, line 216, “[35-36]” should be “[35,36]”

p. 6, line 220-221, “3.3. Toxigenic characteristics of Psychrotolerant and Mesophilic B. cereus Isolates from Chilled tofu” should be “3.3. Toxigenic Characteristics of Mesophilic and Psychrotolerant B. cereus Isolates from Chilled Tofu

p. 6, line 224, “38 mesophilic” please check the number of isolates

p. 6, line 225, “, including psychrotolerant isolates,” should be deleted

p. 6, line 229, the caption of Table 3, “Bacillus cereus” should be “B. cereus

p. 6, Table 3, column 2 and 3, row 1, “Bacillus cereus” should be “B. cereus

p. 7, line 248, “[33,35-36,41]” should be “[33,35,36,41]”

p. 7, line 250, “psychrotolerant and mesophilic” should be “mesophilic and psychrotolerant

p. 7, line 252-253, “3.5. Antibiotic Susceptibility Test of Psychrotolerant and Mesophilic B. cereus Isolates from Chilled tofu” should be “3.5. Antibiotic Susceptibility Test of Mesophilic and Psychrotolerant B. cereus Isolates from Chilled Tofu

p. 7, line 257, “.imipenem” should be “imipenem”

p. 7, line 257, “tetracycline, vancomycin.” should be “tetracycline, and vancomycin.”

p. 7, line 259-260, the legend of Figure 1, “Bacillus cereus” should be “B. cereus

p. 8, line 267, “[42,25]” should be “[25, 42]”.

p.8, line 275-276, “3.6. Biofilm-Forming Capacity at Different Temperature of Psychrotolerant and Mesophilic B. cereus Isolates from Chilled tofu” should be “3.6. Biofilm-Forming Capacity at Different Temperature of Mesophilic and Psychrotolerant B. cereus Isolates from Chilled Tofu”, please check the font size

P. 8, the legend of Figure 2, “Strong” should be “strong”

P. 8, the caption of Figure 2, “Bacillus cereus isolates” should be “B. cereus isolates”

Author Response

# Reviewer 2

Dear reviewer, thank you very much for your kind judgement of our manuscript. We grateful for the time you expended to improve our work. In the following sections, you will find our responses to each of your points and suggestions.

(1) Did the authors check the presence of mesophilic and psychrotolerant B. cereus isolates in the filling water of the packed tofu? Did the authors evaluate the presence or the biofilm formation of mesophilic and psychrotolerant B. cereus isolates on the surface of the packaging material of 20 packaged tofu used in this study?

Thanks for bringing this matter to our attention. We primarily confirmed the B. cereus contamination in the filling water of the packaged tofu and B. cereus did not detected in the filling water of all tofu sample. We did not check the presence of B. cereus on the surface of the packaging material. Our further study will consider the presence of B. cereus on food contact materials in packaged food products.

The result about the detection of B. cereus in filling water inserted in Line 213.

(2) It is necessary to consider and explain the contamination sources of mesophilic and psychrotolerant B. cereus isolates in the tofu manufacturing process.

The microbial quality of tofu may be affected by the handling condition during the tofu manufacturing processing. For instance, the failure of washing step of raw ingredient such as soybean, heating step and cooling step of packed tofu can lead to bacterial growth. Furthermore, improper cleaning or disinfectant procedures for processing line, filling or filling or packing machines, tanks, plastic tray, and cutting knife used during tofu manufacturing process may induce a natural enhancement of the microbial load. Generally, packaged tofu is set in an anaerobic state, and pasteurization-cooling is added as important processing to improve safety quality of tofu. However, foodborne pathogens can still survive due to residual oxygen in headspace by inadequate evacuation or gas flushing, oxygen permeability by inappropriate packaging material, small leakage through poor sealing. After packaging, tofu is preserved and sold refrigerated in retail market or supermarkets under cold condition. Therefore, inadequate temperature control can be main cause for proliferation of psychrotolerant B. cereus. In addition, B. cereus may be capable of existing in spores and might reveal by failure of heating processing such as boiling and pasteurization to eliminate pathogens or by cross-contamination of the surrounding environment.

Above mentioned text included in Line 186-196 and Line 239-243.

(3) There are a number of misspellings and improper word usage in the text.

Abstract, line 11, “B. cereus” should be “Bacillus cereus (B. cereus)”

I revised to ‘Bacillus cereus (B. cereus)’ (Line 11).

 Abstract, line 22, “psychrotolerant and mesophilic” should be “mesophilic and psychrotolerant”

I revised the sentence (Line 20-24)

line 30-33, “the consumption of tofu has increased; therefore,” should be deleted

I revised the sentence like Line 28-30

line 37, “Listeria monocytogens” should be “Listeria monocytogenes (L. monocytogenes)”.

I revised to ‘Listeria monocytogenes (L. monocytogenes)’ (Line 35)

line 39, “[4-5]” should be “[4,5]”

I revised to [4,5] (Line 37)

line 40, “Staphylococcus aureus” should be “Staphylococcus aureus (S. aureus)”

I revised to ‘Staphylococcus aureus (S. aureus)’ (Line 37)

line 40, “[6-7]” should be “[6,7]”

I revised to [6,7] (Line 38)

line 42, “Escherichia coli” should be “Escherichia coli (E. coli)”

I revised to ‘Escherichia coli (E. coli’) (Line 39)

line 43, “Bacillus cereus” should be “Bacillus cereus (B. cereus)”

I revised to ‘Bacillus cereus (B. cereus)’ (Line 41)

line 69, “Biofilm” should be “biofilm”

I revised the sentence to ‘Biofilm cells’ from ‘The cells within Biofilm’ (Line 69) 

line 80-81, “vegetarian vegan and hypocaloric diets” should be “vegetarian, vegan, and hypocaloric diets”

According to comments of other reviewer, the sentence was deleted.  

line 99, “2.2. Microbiological quality of chilled tofu” should be “2.2. Microbiological Quality of Chilled Tofu”

I revised the title of Material and Method section 2.2.

line 109, The incubation temperature and time for MYP agar should be described.

I inserted the information about incubation time and temperature for MYP agar (Line 107-108). 

line 133, “[24-25]” should be “[24,25]”

I revised to [24,25] (Line 134)

line 152-153, “10 µL of over-night culture” should be “An over-night culture of 10 µL”

I revised the sentence like Line 152

line 163, “3.1. Microbiological quality of Chilled Tofu” should be “3.1. Microbiological Quality of Chilled Tofu”

I revised the title of Results and Discussion section 3.1.

line 171-172, the caption of Table 1, “Bacillus cereus, and Escherichia coli in chilled tofu” should be “B. cereus, and E. coli in chilled tofu”

I revised to ‘B. cereus and E. coli’ in the caption of Table 2.

Table 1, column 1, row 4, “Bacillus cereus” should be “B. cereus”

I revised to ‘B. cereus’ (Table 2)

Table 1, column 1, row 5, “Escherichia coli” should be “E. coli”

I revised to ‘E. coli’ (Table 2)

line 180, “Staphylococcus aureus” should be “S. aureus”

I revised to ‘S. aureus’ (Line 197)

line 208, the caption of Table 2, “Bacillus cereus” should be “B. cereus”

I revised to ‘B. cereus’ in the caption of Table 3.

Table 2, column 2, row 1, “Bacillus cereus” should be “B. cereus”

I revised to ‘B. cereus’ (Table 3)

line 180, “could grow at 47 °C, and.” should be “could grow at 47 °C.”

I deleted the word ‘and’ (Line 231).

line 210, “milk, and ice cream” should be “milk and ice cream”

I revised the sentence (Line 231)

line 216, “[35-36]” should be “[35,36]”

I revised to [35,36] (Line 250)

line 220-221, “3.3. Toxigenic characteristics of Psychrotolerant and Mesophilic B. cereus Isolates from Chilled tofu” should be “3.3. Toxigenic Characteristics of Mesophilic and Psychrotolerant B. cereus Isolates from Chilled Tofu”

I revised the title of Results and Discussion section 3.3. 

line 224, “38 mesophilic” please check the number of isolates

I revised the number of mesophilic B. cereus to ‘37’ (Line 260). 

line 225, “, including psychrotolerant isolates,” should be deleted

I deleted the sentence ‘including psychrotolerant isolates’.

line 229, the caption of Table 3, “Bacillus cereus” should be “B. cereus”

I revised to ‘B. cereus’ in the caption of Table 4.

 Table 3, column 2 and 3, row 1, “Bacillus cereus” should be “B. cereus”

I revised to ‘B. cereus’ in Table 4. 

line 248, “[33,35-36,41]” should be “[33,35,36,41]”

I revised the reference number to [33, 35, 36, 41] (Line 282)

line 250, “psychrotolerant and mesophilic” should be “mesophilic and psychrotolerant”

I revised to ‘mesophilic and psychrotolerant’ (Line 284)

line 252-253, “3.5. Antibiotic Susceptibility Test of Psychrotolerant and Mesophilic B. cereus Isolates from Chilled tofu” should be “3.5. Antibiotic Susceptibility Test of Mesophilic and Psychrotolerant B. cereus Isolates from Chilled Tofu”

I revised the title of Results and Discussion section 3.5.

line 257, “.imipenem” should be “imipenem”

I deleted ‘a period’ in front of imipenem (Line 291)

line 257, “tetracycline, vancomycin.” should be “tetracycline, and vancomycin.”

I inserted ‘and’ in sentence (Line 292).

line 259-260, the legend of Figure 1, “Bacillus cereus” should be “B. cereus”

I revised to ‘B. cereus’ in the legend of Figure 1.

line 267, “[42,25]” should be “[25, 42]”.

I revised to [25, 42] (Line 302).

p.8, line 275-276, “3.6. Biofilm-Forming Capacity at Different Temperature of Psychrotolerant and Mesophilic B. cereus Isolates from Chilled tofu” should be “3.6. Biofilm-Forming Capacity at Different Temperature of Mesophilic and Psychrotolerant B. cereus Isolates from Chilled Tofu”, please check the font size

I revised the title to ‘mesophilic and psychrotolerant’ in Results and Discussion section 3.6.

8, the legend of Figure 2, “Strong” should be “strong”

I revised to a small ‘s’ in legend of Figure 2.

the caption of Figure 2, “Bacillus cereus isolates” should be “B. cereus isolates”

I revised to ‘B. cereus’ in the caption of Figure 2.

This manuscript is a resubmission of an earlier submission. The following is a list of the peer review reports and author responses from that submission.